

# Cross species/genera transferability of simple sequence repeat markers, genetic diversity and population structure analysis in gladiolus (*Gladiolus × grandiflorus* L.) genotypes

Varun Hiremath[1], Kanwar Pal Singh[1], Neelu Jain[2], Kishan Swaroop[1], Pradeep Kumar Jain[3], Sapna Panwar[1] and Nivedita Sinha[2]

[1] Division of Floriculture and Landscaping, Indian Agricultural Research Institute, New Delhi, India
[2] Division of Genetics, Indian Agricultural Research Institute, New Delhi, India
[3] National Institute of Plant Biotechnology, New Delhi, India

Corresponding author
Varun Hiremath,
varunhiremath1992@gmail.com

## ABSTRACT

**Background:** Genetic analysis of gladiolus germplasm using simple sequence repeat (SSR) markers is largely missing due to scarce genomic information. Hence, microsatellites identified for related genera or species may be utilized to understand the genetic diversity and assess genetic relationships among cultivated gladiolus varieties.

**Methods:** In the present investigation, we screened 26 genomic SSRs (*Gladiolus palustris, Crocus sativus, Herbertia zebrina, Sysirinchium micranthum*), 14 chloroplast SSRs (*Gladiolus* spp., chloroplast DNA regions) and 25 *Iris* Expressed Sequence Tags (ESTs) derived SSRs across the 84 gladiolus (*Gladiolus × grandiflorus* L.) genotypes. Polymorphic markers detected from amplified SSRs were used to calculate genetic diversity estimates, analyze population structure, cluster analysis and principal coordinate analysis (PCoA).

**Results:** A total of 41 SSRs showed reproducible amplification pattern among the selected gladiolus cultivars. Among these, 17 highly polymorphic SSRs revealed a total of 58 polymorphic alleles ranging from two to six with an average of 3.41 alleles per marker. Polymorphic information content (PIC) values ranged from 0.11 to 0.71 with an average value of 0.48. A total of 4 SSRs were selectively neutral based on the Ewens–Watterson test. Hence, 66.66% of *Gladiolus palustris*, 48% of *Iris* spp. EST, 71.42% of *Crocus sativus* SSRs showed cross-transferability among the gladiolus genotypes. Analysis of genetic structure of 84 gladiolus genotypes revealed two subpopulations; 35 genotypes were assigned to subpopulation 1, 37 to subpopulation 2 and the remaining 12 genotypes could not be attributed to either subpopulation. Analysis of molecular variance indicated maximum variance (53.59%) among individuals within subpopulations, whereas 36.55% of variation among individuals within the total population. The least variation (9.86%) was noticed between two subpopulations. Moderate ($F_{ST}$ = 0.10) genetic differentiation between two subpopulations was observed. The grouping pattern of population structure was consistent with the unweighted pair group method with arithmetic mean (UPGMA) dendrogram based on simple matching dissimilarity coefficient and PCoA.

**Conclusion:** SSR markers from the present study can be utilized for cultivar identification, conservation and sustainable utilization of gladiolus genotypes for crop improvement. Genetic relationships assessed among the genotypes of respective clusters may assist the breeders in selecting desirable parents for crossing.

## INTRODUCTION

Gladiolus (*Gladiolus × grandiflorus* L.) is a commercial bulbous flower cultivated worldwide for its attractive, multi-coloured spikes. *Gladiolus* is one of the largest genera (>265 species) in the family Iridaceae (*Raycheva, Stoyanov & Denev, 2011*). The majority of the wild species are diploid (2n = 30) (*Gladiolus segetum, Gladiolus illyricus, Gladiolus palustris*, and *Gladiolus tristis etc.*), whereas modern cultivars are tetraploids (*Gladiolus × grandiflorus* L.) (*Imanishi, 1989*). Cultivated gladioli are believed to be originated from natural hybridization among number of wild species (*Barnard, 1972*; *Imanishi, 1989*). It is easy to hybridize gladioli owing to their outbreeding nature and high heterozygosity. Hybridization and polyploidy have been greatly responsible for the evolution of gladiolus (*Nemati et al., 2012*). As a consequence of continuous hybridization and selection, gladiolus has been endowed with amazing flower diversity in terms of colour, size, shape and growth habit. Its cut flowers are widely used for decorating vases, bouquet preparation and flower arrangements with huge demand in domestic as well as international markets (*Chaudhary et al., 2018*). Gladiolus ranks fifth in area and production among bulbous flowers cultivated worldwide. In India, it is cultivated in an area of 10.44 thousand hectare with a production of 259.64 thousand MT (*National Horticulture Board, 2021*). Development of novel gladiolus varieties is a continuous process to meet consumer demand in floriculture market. Assessment of genetic variability for superior desirable traits will assist in selection of elite genotypes for crossing programme. Further, understanding the genetic relationship of the parent cultivars would enhance the chances of obtaining new varieties. Also, genetic diversity analysis is a prerequisite for efficient utilization and conservation of existing gladiolus germplasm.

Phenotypic variability of any plant is a result of differences in either DNA sequences or specific genes/modifiers (*De Vicente et al., 2006*). Characterization of a crop germplasm using morphological and physiological traits is not reliable because they are non-abundant and their expression is influenced by environmental changes (*Provan et al., 1999*). Molecular markers circumvent the demerits of non-conventional markers and act as efficient tools to differentiate the closely related genotypes at genotypic level. Characterization of gladiolus cultivars using DNA markers is essential to establish clear distinction between accessions, identification of desirable source for biotic and abiotic stress tolerance, detection of genetic redundancy and in monitoring genetic diversity changes during conservation. In addition, the accurate identification, documentation and conservation of cultivars or breeding lines is very important in order to protect the plant

breeder's rights owing to influx of huge number of new varieties into global markets every year (*Heckenberger et al., 2006*).

Application of molecular markers for germplasm characterization, conservation and crop improvement of gladiolus is very limited despite its popularity. In previous reports, molecular markers such as randomly amplified polymorphic DNA (RAPD) (*Moreno et al., 2011*; *Provan et al., 1999*), inter simple sequence repeats (ISSR) (*Chaudhary et al., 2018*; *Qu et al., 2008*; *Singh et al., 2017b*), sequence characterized amplified region (SCAR) (*Liu et al., 2009*; *Rymer et al., 2010*) and amplified fragment length polymorphism (AFLP) (*Kutlunina, Permyakova & Belyaev, 2017*; *Provan et al., 1999*; *Squirrell et al., 2003*; *Ranjan et al., 2010*) have been utilized for the assessment of genetic diversity and phylogeny of gladiolus species and cultivars. However, more reliable and easily reproducible markers like SSRs are meagre in gladiolus.

Simple sequence repeats have been most preferred markers for genetic studies due to their co-dominant inheritance, high reproducibility, high polymorphism, excellent genome coverage and multi-allelic nature (*Perrier & Jacquemoud-Collet, 2006*). Developing a new set of SSRs for concerned species is a costly affair and consumes more time as it involves sequencing of genomic regions around repeats for design of flanking primers (*Singh et al., 2017c*). Moreover, efficient sequencing requires the intended crop to be diploid, homozygous and with a small genome size. In bulbous flowers, large sequencing efforts (whole genome or a part of genome) for identification of sequence regions flanking simple repeats and development of SSR primers are scarce and genomic information in public databases is lacking or rather very limited (*Krens & Kamo, 2013*). Traditional methods of sequencing enriched libraries for the development of genome-wide SSRs is difficult in cultivated gladiolus due to its large genome size and high heterozygosity. Hence, high closeness due to the conserved genomic regions of crops belonging to the same species or genera may be utilized to study cross-species transferable SSRs. In this manner, SSR markers identified in one plant species can be used directly to study genetic diversity and evolutionary history across closely related species. Heterologous amplification of microsatellites relies on the nucleotide sequence similarity across flanking regions in genome of related species. Therefore, examining cross transferability of SSRs may reduce cost and time required for designing and synthesis of SSRs particularly in species with limited or no genomic information (*Peakall & Smouse, 2012*).

In last few years, microsatellite markers have been developed for iridaceous flowers like *Crocus sativus* (*Nasir et al., 2012*), *Herbertia zebrina* (*Forgiarini et al., 2017*), *Sisyrinchium micranthum* (*Sun et al., 2012*). Recently, a few chloroplast SSRs from plastid sequences of *Gladiolus* (*Rossetto, 2001*), genomic SSRs from *Gladiolus palustris*, an endangered European tetraploid species (*Malkocs et al., 2019*), and expressed sequence tags derived SSRs from *Iris* species were developed (*Takahashi, Yokoi & Takahata, 2016*). Both the chloroplast SSRs and SSR markers derived from ESTs or transcriptome were expected to have high cross species transferability because the earlier had presence in gene-rich regions (*Szczepaniak et al., 2016*) and the latter are located very close to or within functional genes. Despite the ESTs being highly conserved, EST-SSRs often show significant polymorphism among plant species although to a lesser extent than genomic SSRs (*Kalia et al., 2011*).

Cross transferability of SSRs have been investigated in few ornamental plants like *Aspidistra* spp. (*Huang et al., 2014*), cacti (*Bombonato et al., 2019*) *etc*. So far, there are no reported studies on cross-species and cross-genera transferability of genomic or EST derived SSRs in gladiolus. Furthermore, microsatellites information on gladiolus species and related genera available in public databases has not been utilized to assess genetic diversity and characterization of gladiolus genotypes. With these facts, the current study utilized potential microsatellites from related species and genera to detect the extent of cross-transferability as well as to analyze genetic diversity, population structure and infer genetic relationship among gladiolus genotypes.

## MATERIALS AND METHODS

### Plant material and DNA isolation

Plant material consisted a total of 84 Indian and exotic bred gladiolus genotypes collected from different research institutes across India and maintained at research farm of Division of Floriculture and Landscaping, ICAR-Indian Agricultural Research Institute, New Delhi. The details of gladiolus genotypes including parentage, place of collection, flower colour, plant height and flowering time is provided in Table S1. Portions of this text were previously published as part of a preprint (*Hiremath et al., 2021*). Total genomic DNA of individual genotype was isolated from randomly selected young, healthy leaves using a modified cetyltrimethyl ammonium bromide (CTAB) protocol of *Doyle & Doyle (1990)* and further purified to remove excess salts and phenolic residues. For this purpose, $1/10^{th}$ volume of sodium acetate (pH 5.8) and two volumes of absolute alcohol were added to the DNA pellet. The purified genomic DNA was subjected to gel electrophoresis on 0.8% agarose gel stained with ethidium bromide and visualized using a UV transilluminator. DNA quantity was estimated by comparing the band intensities of each sample along with λ DNA (Cat # SD0011; Thermo Fisher Scientific, Waltham, MA, USA). The DNA samples were finally quantified with Nanodrop ND-1000 Spectrophotometer (Nanodrop Technologies Inc, Wilmington, DE, USA). Part of the isolated DNA was diluted with TE buffer to make working concentration of 20 ng/μl and stored in freezer (−20 °C) until further use for polymerase chain reaction (PCR) analysis.

### Source of SSRs and PCR analysis

A total of 65 microsatellite markers identified for gladiolus and related genera/species available in public domain *viz.* cpSSRs (*Rossetto, 2001*), genomic SSRs (*Forgiarini et al., 2017*; *Malkocs et al., 2019*; *Nasir et al., 2012*; *Sun et al., 2012*), EST derived SSRs from *Iris* spp. (*Takahashi, Yokoi & Takahata, 2016*) and intergenic spacer sequences (*Rymer et al., 2010*) were screened for amplification and detection of polymorphism among gladiolus genotypes. All primers were custom synthesized (Invitrogen, Carlsbad, CA, USA) at 100 pmol/μl concentration and stored at −20 °C. Primer working stocks (10 pmol/μl) prepared by adding 5 μl forward and 5 μl reverse primers in 90 μl nuclease free water were stored at 4 °C. Initially, all synthesized primers were screened for amplification using a gradient or a touch-down PCR protocol in few randomly selected gladiolus genotypes to standardize the annealing temperature. PCR amplification was performed in a thermal

cycler with flex gradient technology (peqSTAR®; VWR, Darmstadt, Germany) in 10 μl reaction volume containing 2 μl (20 ng/μl) genomic DNA template, 1 μl of 10X Taq buffer, 1 μl dNTPs (10 mM each), 2 μl of both forward and reverse primer, 0.3 μl 1U *Taq* DNA polymerase (Genei laboratories, Peenya, Bengaluru, India) and 3.7 μl of nuclease free water.

For most of the SSR primers, PCR thermal profile involved initial denaturation at 95 °C for 4 min, followed by 35 cycles of denaturation at 94 °C for 1 min, annealing at 52–60 °C (specific to each primer) for 45 s, extension at 72 °C for 2 min and a final extension at 72 °C for 7 min before cooling down to 4 °C. Touch-down PCR cycling programme was used for few SSRs with the following conditions: initial denaturation at 95 °C for 5 min, followed by 10 cycles of 95 °C for 1 min, annealing (with a touch-down of 59–54 °C, −0.5 °C per cycle for GP 4 & IM 39, 50–45 °C, −0.5 °C per cycle for IM 108 & IM 123, 55–50 °C, −0.5 °C per cycle for IM 112) for 45 s, and 72 °C for 2 min; 25 cycles at 95 °C for 1 min, annealing (54 °C for GP 4 & IM 39, 45 °C for IM 108 & IM 123, 50 °C for IM 112) for 45 s, and 72 °C for 2 min; and a final extension at 72 °C for 7 min before cooling down to 4 °C. Gel electrophoresis was performed to resolve PCR products using 3% agarose SFR® (Super Fine Agarose) in 1× Tris Acetate EDTA (TAE) buffer. The resolved products were visualized under UV light and photographed by using gel documentation system (Syngene, Iselin, NJ, USA). A 100 bp DNA ladder (Fermentas International Inc., Waltham, MA, USA) was used as size standard to determine allele size. The details of primer names, sequences (5′–3′), optimized annealing temperature, allele size for amplified SSRs are given in Table S2.

## Polymorphism detection and genetic diversity analysis

Clearly visible and consistently reproduced PCR fragments for each SSR primer were considered for manual scoring of bands. Alleles were scored as '1' (for presence), '0' (for absence) and '9' (missing data) for a particular band to generate binary data matrix. Total number of alleles for each amplified SSR marker was recorded across all the genotypes. The genotypic data obtained this way was subjected to calculate genetic diversity measures. Polymorphic information content (PIC) for each SSR loci was estimated by determining the frequency of alleles per locus using the formula, $PIC = 1 − \sum (P_i)^2$ where, $P_i$ is the relative frequency of the 'i$^{th}$' allele of a SSR loci (*Perrier & Jacquemoud-Collet, 2006*). The PIC value indicates the genetic variation and also discriminatory power of a marker. Primer resolving power (Rp) was calculated as per the formulae given by *Prevost & Wilkinson (1999)*, $R_p = \sum I_b$, where '$I_b$' is band informativeness = $[1 − \{2(0.5 − p)\}]$ and 'p' is the proportion of the genotypes containing the band. Marker index (MI) for each polymorphic SSR locus was calculated as described by Powell and co-workers (*Perrier & Jacquemoud-Collet, 2006*). The effective multiplex ratio (EMR) was calculated as the number of polymorphic loci present in the germplasm. The percentage of polymorphic bands (PPB) was computed as proportion of total polymorphic bands to total number of bands. Allelic differences at a single locus in a population need to be quantified to measure genetic variation. Therefore, allelic diversity measures viz. observed number of alleles (Na), number of effective alleles (Ne), Shannon's information index (I), observed heterozygosity
(Ho), expected heterozygosity (He) and fixation index (F) were estimated using GenAlEx version 6.5 (*Ohri & Khoshoo, 1983*). Further, Nei's gene diversity (h) and gene flow ($N_m = ((1/F_{st}) - 1)/4$) were estimated using the Popgene v.1.32 software. The Ewens–Watterson Test was performed to check neutrality of each microsatellite loci for whole population used in the study (*Manly, 1985*).

## Population structure analysis

The software programme STRUCTURE v2.3.4 was used to study the underlying population structure among the 84 gladiolus genotypes based on the principle of Bayesian clustering (*Pragya et al., 2010*). The admixture model with correlated allelic frequencies was assumed considering the ancestry of individual genotype in the population. Twenty independent runs were assessed for each fixed ΔK (1 to 10) and each run consisted of 50,000 burn-in length and 100,000 Markov Chain Monte Carlo (MCMC) iterations. The optimum number of subpopulations was identified using STRUCTURE HARVESTER (*Evanno, Regnaut & Goudet, 2005*). Individual genotype was assigned to a subpopulation if at least 70% of its estimated genome fraction value was derived from that group and genotypes with membership probabilities (Q value) less than 0.70 were assigned to a mixed group as admixture.

The SSR genotypic data with individuals assigned to the subpopulations was further used to compute AMOVA (analysis of molecular variance) using GenAlEx software. The genetic relationships among populations were analyzed by computing AMOVA based on allelic frequencies and the number of mutational differences between molecular haplotypes. Allelic patterns across subpopulations depicting number of private alleles, common alleles, and abundant alleles were computed. Pair-wise $F_{st}$ values were estimated for genetic differentiation of subpopulations. The subpopulations were further analyzed on genetic diversity statistics *viz.*, Na, Ne, I, Ho, He, uHe using the GenAlex 6.5.

## Cluster analysis

The 0–1 binary data was utilized to calculate pairwise genetic similarity matrix using Jaccard's coefficient. A radial UPGMA dendrogram was also constructed based on Neighbourhood Joining (NJ) algorithm using simple matching dissimilarity matrix with the help of DARwin 6.0.10 programme (*Peakall et al., 1998*). Robustness of each node of NJ tree was assessed with 5,000-bootstrap replicates. Principal coordinate analysis (PCoA) was performed using GenAlEx v6.5 based on the pair-wise genetic distance matrix between the genotypes and the first two principal coordinates were plotted in two-dimensional space. Mantel test was performed to test the goodness of fit of the similarity matrix generated by genomic, EST and chloroplast derived SSRs (*Mantel, 1967*).

## RESULTS

### PCR analysis and cross transferability

A total of 65 SSRs belonging to gladiolus species and different genera of Iridaceae family were used to amplify DNA from 84 gladiolus genotypes. Genomic SSRs identified for *Crocus sativus* (*Nasir et al., 2012*), *Gladiolus palustris* (*Malkocs et al., 2019*) and EST-SSRs

**Table 1 SSR markers used in the study and their amplification pattern.**

| Sl. No. | Type of marker (species/genera derived from) | No. of screened primers | No. of amplified primers | Amplification % | % of polymorphic markers |
|---|---|---|---|---|---|
| 1 | Chloroplast derived SSRs (*Gladiolus*) | 12 | 12 | 100 | 58.33 |
| 2 | Genomic SSRs (*Gladiolus palustris*, *Crocus sativus*, *Herbertia zebrina*, *Sysirinchium micranthum* & Chloroplast DNA regions) | 28 | 17 | 60.71 | 29.41 |
| 3 | EST derived SSRs (*Iris* spp.) | 25 | 12 | 48.00 | 41.66 |
| | Total | 65 | 41 | | |

identified for *Iris* spp. (*Takahashi, Yokoi & Takahata, 2016*) revealed 71.42%, 66.66% and 48% of cross amplification in all the selected gladiolus genotypes, respectively. However, few genomic SSRs from *Herbertia zebrina* and *Sysirinchium micranthum* could not show any amplification. A total of 41 SSRs produced amplicons in all the gladiolus genotypes. Chloroplast derived SSRs, genomic SSRs and EST-SSRs revealed 100%, 60.71% and 48% amplification in all the gladiolus germplasm, respectively (Table 1). The source of SSR primers developed for different species and their amplification pattern among the gladiolus germplasm is given in Table S3. A total of 17 polymorphic SSRs were obtained which were further utilized in genetic analysis.

## Genetic diversity statistics

Molecular profiling of 84 gladiolus genotypes using 17 polymorphic SSRs revealed a total 58 polymorphic alleles ranging from 2 (G9) to 6 (GP4) with an average of 3.41 alleles per marker. Molecular information generated using 17 polymorphic SSRs is depicted in Table 2. Average polymorphic information content was 0.48 with a range of 0.11 (G9) to 0.71 (G12). Resolving power of primers varied from 1.95 (GP2) to 3.14 (G5) with mean value of 2.48. Marker indices for polymorphic loci diverged from 0.95 (G9) to 2.38 (GP4). Effective multiplex ratio diverged from 2 to 6 and also all SSR loci were 100% polymorphic. Based on Ewens–Watterson test, four non neutral microsatellite loci *viz.* G12, GP7, GP13 and IM31 were detected as their observed homozygosity values lied outside the lower and upper limit of 95% confidence interval (Table S4). Allele wise genetic diversity parameters for all the genotypes are represented in Table S5. Number of effective alleles (Ne) ranged from 1.07 (GP2) to 3.35 (G12) with an average of 2.04 ± 0.2. Average Shannon's information index (I) was 0.76 ± 0.09 with maximum of 1.27 (G12) and minimum of 0.16 (GP2). Observed heterozygosity (Ho) values ranged from 0 (G9) to 0.75 (IM31) whereas expected heterozygosity (He) varied from 0.06 (GP2) to 0.70 (G12) with mean value of 0.43 ± 0.05. Fixation index (F) values ranged from −0.09 (IM31) to 1.00 (G9) with a mean value of 0.24 ± 0.08. Gene flow ($N_m$) for each SSR loci varied from 0.31 (GP2) to 144.84 (G9) with mean value of 54.61 ± 22.02. Values for Nei's genetic diversity (h) extended from 0.06 (GP2) to 0.70 (G12).

**Table 2  Data on allelic diversity generated from 17 polymorphic SSRs in gladiolus.**

| Sl. No | Marker name | Primer (5′–3′) | Source | Tm (°C) | PF | Allele size range (bp) | PIC | R_P | MI | EMR | PPB (%) |
|---|---|---|---|---|---|---|---|---|---|---|---|
| 1 | G5 | F: GCTCACAACAATAATCCTTCCC<br>R: CAATGAACTCAGCAATACCAGC | Singh et al. (2017a) | 60 | 4 | 270–300 | 0.64 | 3.14 | 1.91 | 4 | 100 |
| 2 | G7 | F: GTGTCTTCGGTGCTTTTCTCTT<br>R: CAGCGATAACCTAGAACGAACA | Singh et al. (2017a) | 60 | 4 | 260–320 | 0.29 | 2.00 | 1.91 | 4 | 100 |
| 3 | G8 | F: TCTATGTCAGTGCTCTACCGGA<br>R: GAAGCAAACGAGTCTGTGGAC | Singh et al. (2017a) | 60 | 3 | 280–300 | 0.62 | 2.36 | 1.43 | 3 | 100 |
| 4 | G9 | F: TATAGAGGAATGCGTGTCCGAT<br>R: TACTGCATGACGAGGAAATCAC | Singh et al. (2017a) | 60 | 2 | 400–420 | 0.11 | 2.00 | 0.95 | 2 | 100 |
| 5 | G10 | F: TGCCACTCCAGCATAACTTCTA<br>R: ACTCCTTTTCCTCCCATTCTTC | Singh et al. (2017a) | 60 | 4 | 310–330 | 0.45 | 2.36 | 1.91 | 4 | 100 |
| 6 | G11 | F: AAAGTCCCTCCTCTCCTCTGAT<br>R: GAGCTTGTTACTGAACGGAACC | Singh et al. (2017a) | 60 | 3 | 480–510 | 0.64 | 2.60 | 1.43 | 3 | 100 |
| 7 | G12 | F: GGCATCCTTCCTCTCCGT<br>R: CGGCCTTGGGTGTAGAAGTAG | Singh et al. (2017a) | 60 | 4 | 200–240 | 0.71 | 2.79 | 1.91 | 4 | 100 |
| 8 | GP 2 | F: TTGTTACTGGTGCGGACTCC<br>R: CAGGTCCGATTGCTTGAGGA | Malkocs et al. (2019) | 58 | 3 | 210–270 | 0.12 | 1.95 | 1.43 | 3 | 100 |
| 9 | GP 4 | F: ATGCCTTTGTCCTCTCACCT<br>R: TTTGTCCCTAATTGGAACACGTC | Malkocs et al. (2019) | 54 | 6 | 180–310 | 0.52 | 2.17 | 2.38 | 5 | 100 |
| 10 | GP 7 | F: CCAAGTAAGTGATGGCGGC<br>R: GGGTCTAGAGAAGGCTTGGG | Malkocs et al. (2019) | 56 | 3 | 190–210 | 0.66 | 2.62 | 1.44 | 3 | 100 |
| 11 | GP 13 | F: AAACCCTCACTTCGGAGATCA<br>R: TAAAGTCAGTCAGCTGTAACACTG | Malkocs et al. (2019) | 54 | 3 | 280–300 | 0.66 | 2.67 | 1.44 | 3 | 100 |
| 12 | GP 15 | F: GGGTCATCGCCTGTCATGAA<br>R: TCGTATCGGCTTGTTGGCTG | Malkocs et al. (2019) | 54 | 4 | 190–210 | 0.68 | 3.05 | 1.91 | 4 | 100 |
| 13 | IM 31 | F: AAGCAAAAGGTTTTCCATTCC<br>R: GTTTCTTGTCGAGGAACATGC | Tang et al. (2009) | 52 | 4 | 300–420 | 0.69 | 2.90 | 1.91 | 4 | 100 |
| 14 | IM 86 | F: GGGTTTGTATTGTTTGTTGGAGA<br>R: GGGTGATGTGGTCCTTGTAGA | Tang et al. (2009) | 60 | 2 | 200–220 | 0.21 | 2.00 | 0.96 | 2 | 100 |
| 15 | IM 108 | F: TCTCCTCCTGTCCGTCTATCC<br>R: AGTCGTCCAAATCTCCGAACT | Tang et al. (2009) | 45 | 2 | 320–390 | 0.45 | 2.43 | 0.96 | 2 | 100 |
| 16 | IM 39 | F: CCCTAGCAAACATCTCTTCCA<br>R: TGTTATCAGCAAGCAGTCCAG | Tang et al. (2009) | 54 | 3 | 380–500 | 0.25 | 2.00 | 1.44 | 3 | 100 |
| 17 | IM 224 | F: AGAGAAGAGAGCATGGCGATA<br>R: GCGAGAAGTGGCATAAAGAGA | Tang et al. (2009) | 46 | 4 | 200–280 | 0.41 | 2.07 | 1.91 | 4 | 100 |
| Total | | | | | 58 | | 8.11 | 42.10 | 27.27 | | |
| Average | | | | | 3.41 | | 0.48 | 2.48 | 1.60 | | |

**Note:**
PF, Polymorphic fragments; PIC, Polymorphic information content; R_P, Resolving power; MI, Marker index; EMR, Effective multiple ratio; PPB (%), The percentage of polymorphic bands.

## Population structure and cluster analysis

Population structure of gladiolus genotypes was analyzed using Bayesian model approach. All the genotypes were assigned to two distinct subgroups based on maximum likelihood and delta K value (ΔK = 2) by their inferred genome fraction value (Fig. 1). The structure
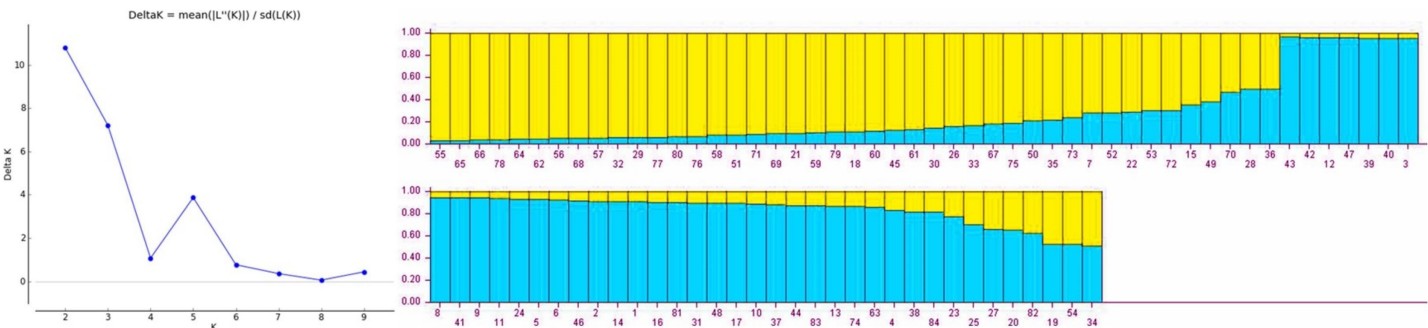

**Figure 1 Estimated population structure of 84 gladiolus genotypes as revealed by 17 polymorphic SSR markers for ΔK = 2 from an assumed range of 1–10 based on Evanno method.** Blue color indicates sub-population 1, yellow color indicates sub-population 2.

depicted two sub-populations (S1 and S2) composed of gladiolus genotypes studied (Fig. 1). Out of 84 genotypes, 72 were assigned to two sub groups and 12 were retained in the mixed group as admixture. Sub-population 1 (green group) consisted of 35 genotypes representing 41.66%, whereas sub-population 2 (red group) contained 37 genotypes representing 44.04% of the total number of the genotypes in the study. Analysis of molecular variance and F statistics differentiating subpopulations is presented in Table S7. Maximum variance (53.59%) was revealed among individuals within subpopulations whereas 36.55% of variation observed among individuals within total population. However, 9.86% variation was noticed between two subpopulations. Fixation indices including $F_{ST}$, $F_{IS}$ and $F_{IT}$ values were 0.10, 0.41 and 0.46. Subpopulation 1 had highest average genetic diversity parameters than subpopulation 2 *viz*. Na (3.12 ± 0.26), Ne (1.99 ± 0.21), I (0.73 ± 0.10), Ho (0.34 ± 0.05) and He (0.41 ± 0.06) (Table S8). However, highest F (0.22 ± 0.09) value and polymorphic loci % (100.00%) was observed in subpopulation 2.

Radial UPGMA dendrogram created based on simple matching dissimilarity matrix differentiated 84 genotypes into two distinct major clusters with 42 genotypes each (Fig. 2). Composition of Cluster I and II was quite similar to the composition of subpopulation 1 and 2, respectively (Table S6). Coefficients of dissimilarity ranged from 0.16 to 0.89 with an average value 0.48. Punjab Lemon Delight and Vicki Lin (0.89) had highest degree of dissimilarity while least value was observed between Pusa Archana and Pusa Bindiya (0.16). Principal coordinate analysis also depicted similar groupings as in population structure and UPGMA cluster. Principal coordinate 2 explained maximum variance of 21.28% followed by first principal coordinate with variance of 11.64%. Although the marker effectiveness indicated goodness of fit of three SSR marker types, there was no Mantel correlation between genomic, chloroplast and EST-SSRs.

## DISCUSSION

### Cross transferability and PCR amplification

Molecular markers facilitate precise and quick varietal identification, germplasm characterization and conservation. Microsatellite markers have been the most preferred in molecular studies because of their codominance, high discrimination power, multiallelic

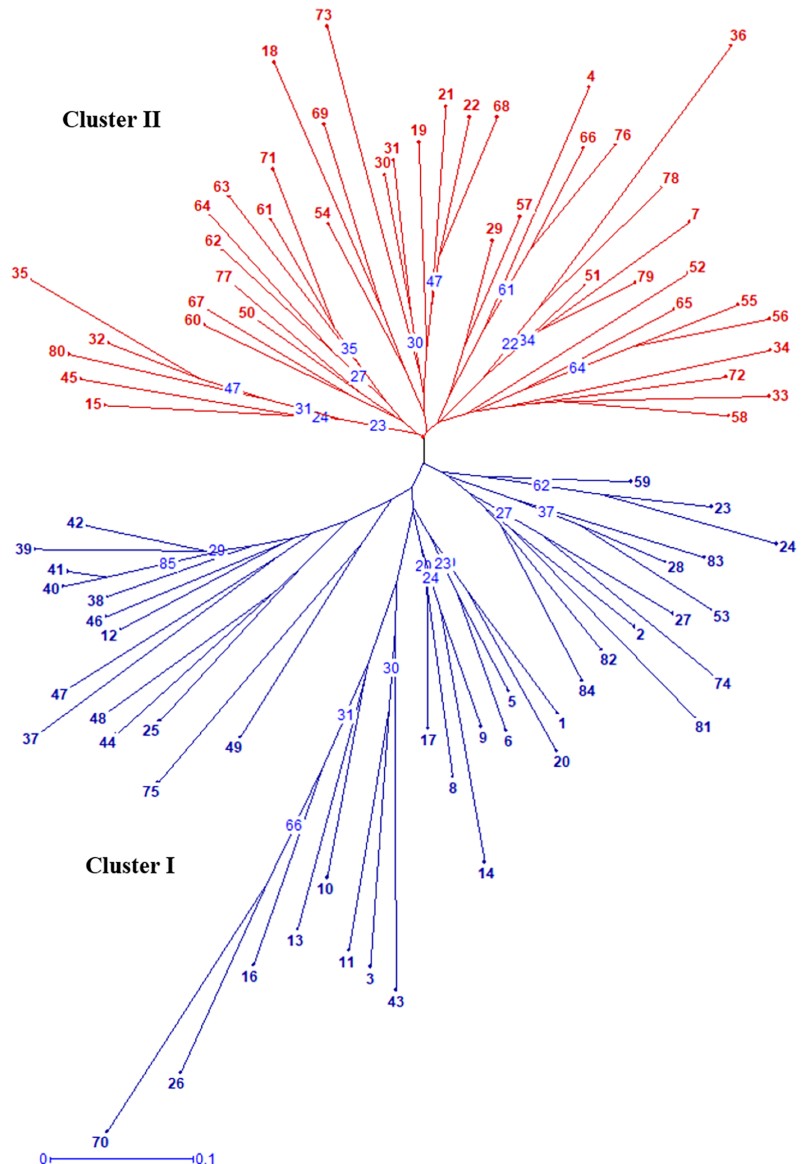

**Figure 2 Radial UPGMA dendrogram of 84 gladiolus genotypes constructed based on simple matching dissimilarity coefficient matrix using 17 polymorphic SSRs.** The numerical values representing the different genotypes are the genotype ID given in Table S5. The numbers in blue within tree nodes represent respective bootstrap values.

nature and cross transferability within genera/family (*Perrier & Jacquemoud-Collet, 2006*). They are easy to use, cost effective and amenable to automation as compared to other markers. Development of novel SSRs for concerned species involves sequencing of flanking genomic regions around simple repeats and is a costly affair in the absence of genomic information (*Singh et al., 2017c*). In geophytes, particularly in gladiolus, it is comparatively difficult to achieve whole or part of genome sequence due to high heterozygosity, polyploidy and huge genome size (*Krens & Kamo, 2013*). Under such circumstances, a feasible strategy to detect microsatellite loci for a target species is through cross-species and cross-genera transferability of SSRs (*Mantel, 1967*). In the present investigation, 17 highly

polymorphic SSRs were detected from a set of 65 microsatellites reported within iridaceae family members. Seven chloroplast SSRs of gladiolus, five genomic SSRs of *Gladiolus palustris* and five EST derived SSRs of *Iris* could successfully amplify in gladiolus suggesting transferability due to relatedness in cultivars. Chloroplast and genomic SSRs have been widely used to study phylogenetic evolution of plants in recent years (*Singh et al., 2017a*; *Squirrell et al., 2003*). It is established that chloroplast genome is characterized by conserved genic sequences, non-recombination and maternal inheritance in plants (*Provan et al., 1997*). Chloroplast SSRs revealed higher level of diversity in rice and barley species in contrast to chloroplast derived RFLPs (*Pritchard, Stephens & Donnelly, 2000*; *Provan, Powell & Hollingsworth, 2001*). Further, the microsatellites reported for *Gladiolus palustris* showed positive cross species transferability (66.66%) in gladiolus cultivars. This indicated the presence of conserved genomic regions between *Gladiolus palustris* and modern gladiolus cultivars. Cross species amplification of these microsatellites has been also studied in individuals of *Gladiolus imbricatus* and *Gladiolus tenuis* (*Malkocs et al., 2019*). In support of our findings, close genetic relationship between modern gladiolus cultivars and *Gladiolus palustris* was revealed using chloroplast DNA regions (*Singh et al., 2017a*). In addition, *Iris* EST-SSRs also portrayed successful cross genera amplification (48%) in gladiolus. Our findings are consistent with the prediction that cross-transferability of SSRs can vary from 50% to 100% within a genus, while transferability across genera is generally less than 50% (*Peakall & Smouse, 2012*). SSR markers derived from ESTs or transcriptome have high rate of cross transferability as they are highly conserved, located very close to or within functional genes (*Kalia et al., 2011*). The possibility of cross transferability is high when the repeat sequences and flanking region consisting selected primer region is conserved across taxa, although the polymorphism generated may be less. A number of cross-transferable monomorphic markers observed in this study may be attributed to the same fact. It was also seen that mantel correlation was negligible (<0.06) between genomic, chloroplast derived and EST-SSRs that could be ascribed to highly conserved nature of gladiolus chloroplast-derived and *Iris* EST-SSR markers and more diverse nature of genomic SSRs indicating genetically diverse grouping patterns for the three SSR types. In a similar line of study, *Debener (2012)* analyzed genetic diversity of *Euphorbia pulcherrima* accessions using EST-SSRs developed for *Euphorbia esula* through cross species amplification. In ornamentals, congeneric transferability of microsatellite markers has been investigated in *Iris* spp. (*Singh et al., 2018*), *Aspidistra* spp. (*Huang et al., 2014*) and cacti (*Bombonato et al., 2019*).

## Marker efficiency and allelic diversity measures

Molecular profiling of 84 gladiolus genotypes using 17 polymorphic SSRs revealed that at least nine markers were highly informative (PIC > 0.5). Representative gel electrophoresis profiles for markers G5 and GP13 are presented in Fig. 3. Abundant polymorphism and more genomic coverage of molecular markers increases accuracy in genetic diversity studies (*Perrier & Jacquemoud-Collet, 2006*). This reduces the amount of genotyping required for phylogenetic analysis of crop plants. According to *Botstein et al. (1980)*, a

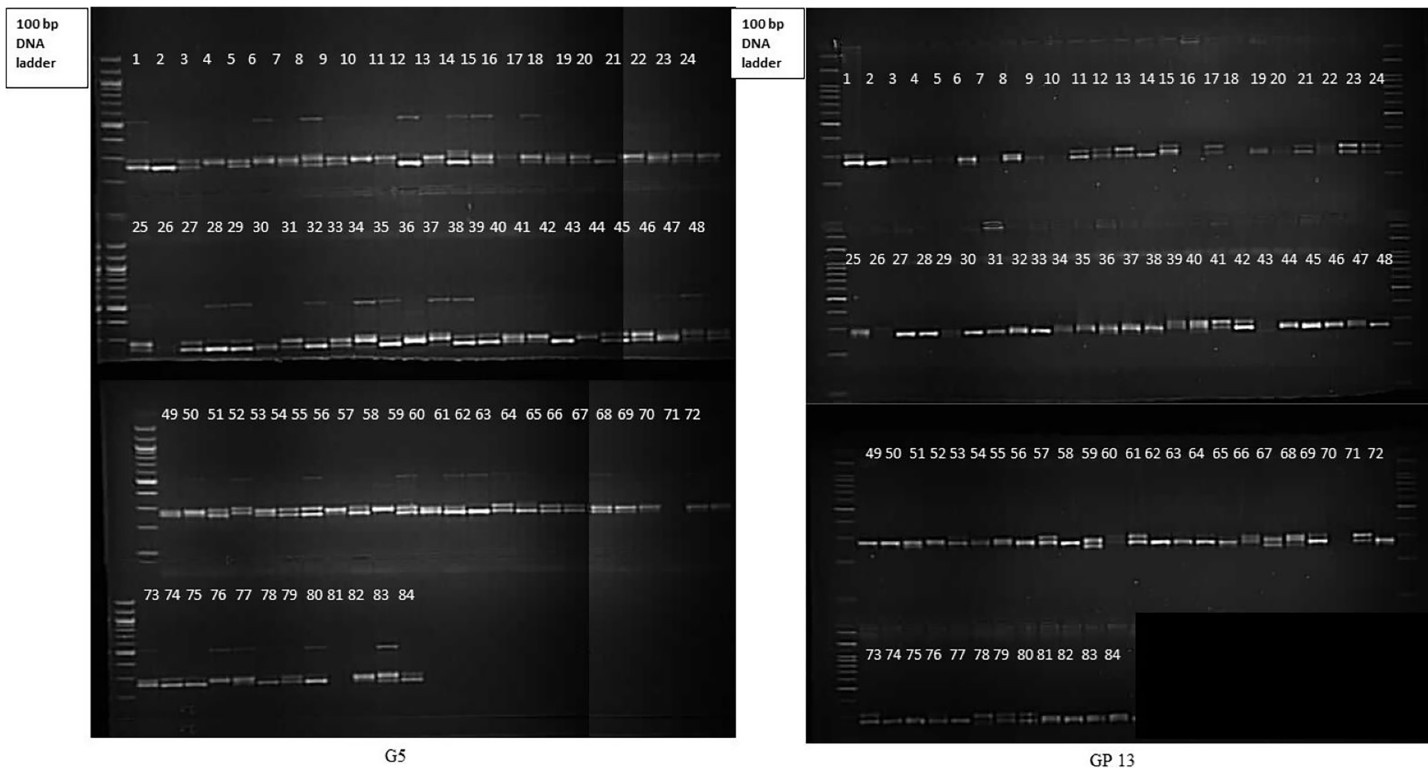

**Figure 3 Gel electrophoresis profile of 84 gladiolus genotypes revealed by G5 and GP13.**

DNA marker with PIC value more than 0.5 is said to be highly informative. PIC and Rp (>0.50 and >2.0, respectively) values for most of the loci suggested that SSRs were efficient enough to distinguish gladiolus germplasm in the study. Higher PIC value of SSRs may be attributed to their codominant and multi-allelic nature. Marker indices define total utility of marker system in estimating genetic variation within germplasm pool (*Chaudhary et al., 2018*; *Rymer et al., 2010*). GP4 had highest marker index value of 2.38. Estimates of PIC, RP and MI signify the overall ability of such markers in detecting genetic variation and infer genetic relationships between accessions (*Powell, Machray & Provan, 1996a*). In the current study, SSR markers showed comparatively higher polymorphism against those DNA markers used in earlier reports (*Provan et al., 1999*; *Moreno et al., 2011*; *Chaudhary et al., 2018*; *Singh et al., 2017b*).

Allele—wise mean genetic diversity parameters indicated effectiveness of these SSRs to characterize the gladiolus germplasm used in the present study. Average h (equivalent to average He) indicated higher frequency of heterozygotes at single locus when chosen randomly. Average He was comparatively higher than Ho for tested SSRs. Inbreeding coefficient values for most of the SSRs were low suggesting less fixation of alleles. Average $N_m$ revealed high allelic diversity among gladiolus genotypes. Similarly, allelic diversity has been estimated using EST-SSRs in *Iris* spp. (*Takahashi, Yokoi & Takahata, 2016*; *Singh et al., 2018*; *Tacuatiá et al., 2012*; *Tang et al., 2018*) and genomic SSRs in *Crocus* (*Nasir et al., 2012*), *Herbertia* (*Forgiarini et al., 2017*) and *Gladiolus* (*Malkocs et al., 2019*). Non neutral markers indicated their possible linkage to phenotypic traits or genes under
selection. *Kirk & Freeland (2011)* suggested that non neutral markers may show unusual genetic divergence for traits under selection.

## Population structure and cluster analysis

STRUCTURE analysis based on Bayesian approach grouped the overall gladiolus germplasm into two subpopulations however, no clear distinction among the Indian and exotic germplasm was established. Exotic genotypes could not be demarcated from Indian bred genotypes because introduced exotic gladioli have been used as parents in crossing programme and therefore, fall within pedigree of Indian genotypes (Table S1). It is known that no gladiolus species is native to India and it was introduced to India in later part of 19[th] century. Cultivated gladioli (*Gladiolus × grandiflorus*) are complex hybrids and have been evolved from interspecific hybridization among wild species viz. *Gladiolus cardinalis, Gladiolus daleni, Gladiolus oppositiflorus, Gladiolus papilio, Gladiolus carneus, Gladiolus cruentus, Gladiolus tristis*, and *Gladiolus saundersii* (*Huxley, Griffiths & Levy, 1992*). This is evident from the results where 12 out of 84 genotypes shared less than 70% of its estimated genome fraction with either of the two subpopulations and thus revealed mixed populations. In a similar line of study, *Chaudhary et al. (2018)* revealed presence of mixed population among gladiolus germplasm while analyzing STRUCTURE of 53 gladiolus genotypes using ISSR markers. AMOVA results showed minor variation between two subpopulations whereas individuals within total population depicted maximum variance. Wright's F statistics differentiated whole germplasm into two subdivided population based on level of allele frequencies shared among individuals. According to *De Vicente, Lopez & Fulton (2004)*, if $F_{ST}$ value is within the range of 0.05–0.15, then subpopulations are assumed to be moderately differentiated at genetic level (*De Vicente et al., 2006*). Moderate ($F_{ST} = 0.10$) genetic differentiation of allelic frequencies was observed among two subpopulations. The average heterozygotes in each subpopulation ($F_{IT} = 0.41$) and among subpopulations ($F_{IT} = 0.46$) indicated existence of large non-random mating among individuals within subpopulations. This may be attributed to complex ploidy level, high heterozygosity and cross pollination nature of *Gladiolus*. Various factors including pollen movement, germplasm exchange, natural selection, reproduction system and geographical distribution decide the level of variability existing among the populations. Subpopulation 1 was genetically more diverse than subpopulation 2 accounting highest average genetic diversity estimates.

## Genetic relationships

In our study, the grouping pattern arising from STRUCTURE were consistent with those obtained from UPGMA dendrogram (Fig. 2) and PCoA (Fig. 4) in terms of genotype number and composition of clusters excluding few admixtures. A detailed insight into genetic relationship among gladiolus varieties was inferred from polymorphic SSR data based on Jaccard's similarity matrix and UPGMA clustering. Two major clusters containing 42 genotypes each were identified at an average pairwise similarity of 0.49. Highest degree of similarity for cultivar pairs viz. Dhanvantari—Fire Flame (Jaccard's coefficient = 0.97) followed by Yellow Star—Neel Rekha (Jaccard's coefficient = 0.96) was

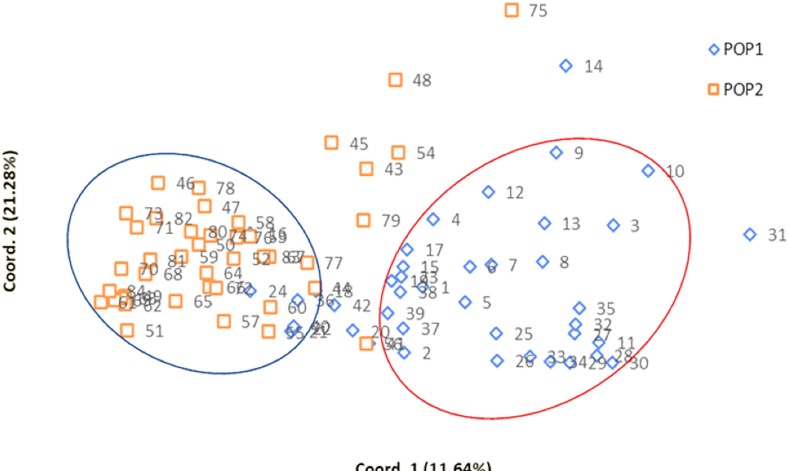

**Figure 4 Two dimensional PCoA scatter plot showing composition of two clusters.** Bullets in blue with numbers represent genotypes of subpopulation 1 whereas bullets in orange with numbers indicate genotypes of subpopulation 2 (Table S5).

recorded and it was evident from their close presence in Cluster II, while lowest degree of similarity between cultivar pairs viz. Punjab Dawn—Punjab Lemon Delight and Arka Arti—Punjab Lemon Delight was observed and thus their presence in different clusters could be noticed. Based on simple matching dissimilarity matrix, Punjab Lemon Delight—Vicki Lin (coefficient of dissimilarity = 0.89) had highest degree of dissimilarity. Presence of Punjab Lemon Delight—Vicki Lin in cluster I and II respectively, could be justified to their clustering pattern. Punjab Lemon Delight and Vicki Lin were found to be genetically distinct among gladiolus genotypes used in the study.

Cultivar pairs which shared common parent/s (either male/female or both) in their derivation were found closely spaced in cluster I. Indian bred cultivars namely Arka Amar (Watermelon Pink × Arka Aarti), Arka Darshan (Watermelon Pink × Shirley), Arka Sapna (Green Woodpecker × Friendship), Pusa Chandni (Green Woodpecker × White Butterfly), Punjab Glad 1 (Happy End × True Yellow) and Punjab Glance (Happy End × Yellow Stone) were grouped together owing to common female parent during hybridization. While Punjab Pink Elegance (Suchitra × White Prosperity) and Punjab Lemon Delight (Jacksonville Gold × White Prosperity) were noticed in the same cluster showing similarity in their male parentage. Thus, presence of Suchitra, Jacksonville Gold and Yellow Stone in the same group could be highly obvious. Occurrence of Pusa Sarang (OP seedling obtained from 'White Oak'), Pusa Shagun (White Oak × Oscar), Pusa Mohini (Ave × Christian Jane) and Pusa Kiran (OP seedling obtained from 'Ave') in cluster I was evident showing their affinity towards parentage. However, rest of the genotypes could not share any common parentage but clustered together. This might be attributed to highly heterozygous nature of crop.

Correspondingly, cluster II was also formed by the genotypes sharing common lineage. Presence of Punjab Dawn (Suchitra × Melody), Arka Arti (Shirley × Melody), Amethyst

(Lavender Puff × Tropic Sea), Neelima (Snow Princess × Tropic Sea), Arka Naveen (74–39-1 × Tropic Sea), Pusa Sringarika (Mayur × Heady Wine) and Pusa Urmi (Berlew × Heady Wine) in cluster II might be attributed to owing to their common male parentage during crossing. Similarly, Punjab Morning (Sancerre × White Prosperity) and Anjali (Sancerre × Rose Spire) were placed close together with female parent 'Sancerre'. Malaviya Shatabdi, an induced mutant of Punjab Dawn was also present in this cluster. Kalima and Pusa Suhagin were open pollinated seedlings of Sylvia and found related to each other in this cluster. Even though Pusa Srijana and Pusa Urmi had common parentage *i.e.*, Berlew × Heady Wine, however, their presence in cluster I and II, respectively might be due to chance similarity in their parentage at genotypic level. Few genotypes that did not share any common parentage were also grouped together in both the clusters because of greater degree of similarity in genetic constitution of ancestors. It was also suggested that chloroplast SSRs or EST derived SSRs may not differentiate the related species and cultivars due to conserved genome or positions within genic regions in the genome of ancestors. However, pedigree information was not available for some of the genotypes as it is essential for comparative analysis with SSR profiles. In general, our results were more consistent with clustering pattern obtained based on AFLP (*Provan et al., 1999*), RAPD (*Powell et al., 1996b*) and ISSR (*Chaudhary et al., 2018*) data. In a similar study, Singh and co-workers described phylogenetic relationship among gladiolus cultivars using sequenced chloroplast DNA regions (*Singh et al., 2017a*).

## CONCLUSIONS

To conclude our study, we are first to report cross transferability of SSRs developed for *Gladiolus palustris* and *Iris* spp. to analyze genetic diversity, population structure and genetic relationships among cultivated/modern gladiolus genotypes. Microsatellite markers detected in the current study have great discriminatory power and highly informative to study genetic diversity and molecular characterization of gladiolus germplasm. However, genetic variability obtained in the STRUCTURE of gladiolus germplasm was narrow indicating use of limited gene pool in breeding new varieties. Further, genetic relationships assessed among gladiolus genotypes will assist the breeders to select desirable parents for hybridization. Identified SSRs will be helpful for identification, documentation and conservation of gladiolus varieties and also can be very useful in marker assisted breeding programme. These markers also help in protection against unauthorized commercialization of varieties and protection of plant breeder rights.

### Funding

The authors received no specific funding for this work. Varun Hiremath received a Senior Research Fellowship for his Doctoral research work from the ICAR-Indian Agricultural Research Institute, New Delhi. Varun Hiremath also received laboratory facilities from the Wheat Molecular Biology and Biotechnology Lab, Division of Genetics, ICAR-IARI.

The funders had no role in study design, data collection and analysis, decision to publish, or preparation of the manuscript.

### Grant Disclosures
The following grant information was disclosed by the authors:
ICAR-Indian Agricultural Research Institute, New Delhi.
Wheat Molecular Biology and Biotechnology Lab, Division of Genetics, ICAR-IARI.

### Competing Interests
The authors declare that they have no competing interests.

### Author Contributions
- Varun Hiremath conceived and designed the experiments, performed the experiments, analyzed the data, prepared figures and/or tables, authored or reviewed drafts of the article, and approved the final draft.
- Kanwar Pal Singh conceived and designed the experiments, authored or reviewed drafts of the article, and approved the final draft.
- Neelu Jain analyzed the data, authored or reviewed drafts of the article, and approved the final draft.
- Kishan Swaroop conceived and designed the experiments, authored or reviewed drafts of the article, and approved the final draft.
- Pradeep Kumar Jain analyzed the data, authored or reviewed drafts of the article, and approved the final draft.
- Sapna Panwar analyzed the data, prepared figures and/or tables, and approved the final draft.
- Nivedita Sinha analyzed the data, prepared figures and/or tables, and approved the final draft.

### Data Availability
 The raw data is available in the Supplemental Files.

### Supplemental Information
Supplemental information for this article can be found online at http://dx.doi.org/10.7717/peerj.15820#supplemental-information.

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
