# Peer review of "Cross species/genera transferability of simple sequence repeat markers, genetic diversity and population structure analysis in gladiolus (Gladiolus × grandiflorus L.) genotypes"

_PeerJ, doi:10.7717/peerj.15820_

## Round 0.1 · original submission · Major Revisions

Dear Dr. Hiremeth

The reviewers have suggested a suite of major and minor changes to improve the quality of your manuscript. In addition to the reviewers' comments, you will also have to address the following concerns:

1. In the background section of the abstract, authors mention about the evolutionary history of gladiolus; however, this has not been presented in the Results and Discussion parts. Accordingly, either the background section needs to be amended or the Results/Discussion modified to ensure coherence.

2. In the Methods section of the Abstract, the authors mention about Principal Component Analysis, but subsequently they have mentioned the results of Principal Coordinate Analysis. This needs to be checked and corrected.

3. Conclusions section is missing from the Abstract, it should be added alongside a brief mention of the future line of research.

4. Vague statements need to avoided unless absolutely necessary and beyond any doubt; as in line 40 'the most diverse family'.

5. The sentences should not begin with abbreviations as in line 68 (SSRs). Please check and apply this to the whole manuscript.

6. Please ensure that words are not joined with each other as in lines 116 (at110) and 117.

7. The Conclusions section needs to be added at the end of the manuscript, summarizing the major findings and future perspectives.

In light of these comments, you have to thoroughly revise the manuscript and submit within the stipulated time.

Reviewer 1 ·

Basic reporting

no comments

Experimental design

No comments

Validity of the findings

no comments

Additional comments

The manuscript (MS) focuses on the investigation of Cross species/genera transferability of SSR markers and genetic diversity and population structure study of gladiolus genotypes using SSR markers. The work looks good but there are certain issues which need to be addressed before the MS is approved for publication. The comments are listed below.

L-25. Write “four” as “4” as it is always good to maintain uniformity throughout the MS.
L-28. What could be reason why 12 genotypes have mixed clustering? Incorporate the explanation in the discussion part of the MS. Also avoid using “assigned” repeatedly. Use another synonym of the word.
L-55. Put a reference after “ Changes”.
L-116. Put a space between “at” and “100”.
L-148. “needs” to be changed to “need”.
L 164- 167. Simplify the sentences.
L-171. Change “subject” to “subjected”.
- Mantel teat could be performed for 3 groups of SSRs obtained from Crocus sativus, Gladioles patustris and EST-SSR to identify maker correlation and effectiveness.
- Mantel test for chloroplast derived SSRs, genomic SSRs and EST-SSR should be conducted to determine the correlation ship and efficiency of 3 SSR marker types.
L 196-202. The authors should avoid repeated used of “ranged” in these sentences. Use different words like “ varied from” “ extended from” etc. The use of same words repeatedly does not look good and make the sentences monotonous.
- The authors are also suggested to determine EMR and PPB of the primers to represent more accurate information of the primer effectiveness.
L-208. Bring Fig,S1 as main figure along with structure diagram.
L-226-228. Avoid using repeated use of “dissimilarity”.
L-230. Bring Fig.S2 as main Figure.
- The gladioles are known to be highly diverse genetically according to the authors. The authors should explain why the 84 genotypes grouped only to two subpopulations in spite of its rich diverseness. The explanation should be incorporated in the discussion part of the MS.
L-238. Put a reference after “family”.
L-241. Put a reference after “information”.
L-269. Put a reference after “studies”.
L-280. Change “high” to “higher”.
- There are several minor grammatical errors and spelling mistakes in different parts of the MS which need to be corrected and should be highlighted so that modifications may be identified for next revision.

Annotated reviews are not available for download in order to protect the identity of reviewers who chose to remain anonymous.

Reviewer 2 ·

Basic reporting

No comment

Experimental design

No comment

Validity of the findings

No comment

Additional comments

Comment for the author:
The work cannot be accepted in its current state. It should be revised before being submitted again for revision.
 Kindly mentioned one para in introduction section about the market worth, demands in different sector along with annual growth rate in order to reflect the importance of work.
 Include the company names and the nation from where the equipments were purchased. Whenever applicable.
 Some where the genera name in MS is not italic.
 Most of the references are too old. Cite latest references.
 Author contribution, funding source, Declaration such as competing /conflict of interest, Consent for publication, Originality of Research content, Data availability etc are not mentioned in Main MS. Kindly check.

Reviewer 3 ·

Basic reporting

Is the order of references in the text of the manuscript correct? Please, check.

Experimental design

No comment.

Validity of the findings

No comment.

Additional comments

Whereas gladiolus as economically important flowering plants reflecting the wide range of morphological variability of the inflorescences, the aim of current gladiolus breeders is to create interesting new genotypes carrying a variety of colour, size, texture and shapes of flowers and inflorescences. Therefore the genomic screening of gladiolus germplasm by molecular markers can provide useful and valuable information.
The methodology of the experiments is well defined. The information needed to ensure reproducibility of experiments is well defined. The polymorphism analysis is correctly evaluated and interpreted.
The authors compiled a comprehensive diversity study of a collection of gladiolus genotypes, which provides practical data.

---

## Round 0.2 · accepted · Accept

Dear Dr. Hiremath

Thank you for thoroughly revising the manuscript as per the reviewers' comments.

This is to inform you that your manuscript - Cross species/genera transferability of simple sequence repeat markers, genetic diversity and population structure analysis in gladiolus (Gladiolus × grandiflorus L.) genotypes - has been Accepted for publication.

Congratulations!

Reviewer 2 ·

Basic reporting

The author has solved the comments raised on the MS. I would recommend to accept and finally publish the MS in this Journal.

Experimental design

Fine

Validity of the findings

Fine